# Effects of Robotic Interactive Gait Training Combined with Virtual Reality and Augmented Reality on Balance, Gross Motor Function, Gait Kinetic, and Kinematic Characteristics in Angelman Syndrome: A Case Report

**DOI:** 10.3390/children9040544

**Published:** 2022-04-12

**Authors:** Sangkeun Han, Chanhee Park, Joshua (Sung) H. You

**Affiliations:** 1Sports Movement Artificial-Intelligence Robotics Technology (SMART) Institute, Department of Physical Therapy, Yonsei University, Wonju 26493, Korea; lagoestrella@gmail.com (S.H.); chaneesm@gmail.com (C.P.); 2Department of Physical Therapy, Yonsei University, Wonju 26493, Korea

**Keywords:** Walkbot, Angelman, robotic-assisted gait training, kinematics, kinetics

## Abstract

Angelman syndrome (AS) is a genetic neurological disorder resulting in cognitive and neuromuscular impairments, such as lack of safety awareness and attention, as well as lack of balance and locomotor control. The robotic interactive gait training (RIGT) system is designed to provide accurate proprioceptive, kinematic, and kinetic feedback, and facilitate virtual reality and augmented reality (VR-AR) interactive exercises during gait training. In the present case report, we examined the effect of an innovative hip-knee-ankle interlimb-coordinated RIGT system. We utilized this therapeutic modality in a participant with Angelman syndrome (AS). Gross motor function measures, risk of fall, and gait-related kinetic (force), and kinematic (joint angle) biomechanical characteristics were assessed before and after 20 sessions of RIGT with VR-AR. We found RIGT with VR-AT improved gait ability, as shown by Performance-Oriented Mobility Assessment score, gross motor function by Gross Motor Function Measure score, balance by Pediatric Balance Scale score, knee and hip joint kinetics, and kinematics during gait. Our clinical and biomechanical evidence provide important clinical insights to improve the effectiveness of current neurorehabilitation approaches for treating patients with AS in balance and locomotor control and reduce the risk of falling.

## 1. Introduction

Angelman syndrome (AS) is a genetic neurological disorder resulting in cognitive and neuromuscular impairments, such as lack of safety awareness and attention, as well as lack of balance and locomotor control. This may predispose patients to a high risk of fatal fall injuries [1,2,3]. In particular, such impaired balance and locomotor control are complicated by force and kinematic coordination, which affect single-limb stance balance or stability and also compromise spatiotemporal variables during gait [4]. Specifically, an abrupt or uncoordinated waddling gait pattern, with reduced stride length, variable cadence and gait speed, and increased double-support and stance-phase duration were commonly observed in the gait of children with AS [1,5]. Furthermore, the lack of motivation and interest or easy distraction during treatment activities are identified as barriers to rehabilitation. However, conventional physical therapy has difficulty in increasing the motivation and interest levels of children with AS [3,4]. Therefore, an effective and sustainable intervention is needed while considering balance and locomotor coordination, and factors affecting personal motivation and interest level [6].

To mitigate balance and locomotor control impairments in AS, conventional physical therapy, including a body-weight-supported treadmill (BWST) exercise, was used but failed to yield beneficial effects and lacked any supporting evidence [7,8]. Lowe et al. reported that the BWST exercise did not significantly improve Gross Motor Function Measure (GMFM) score (8.6%) in 12 participants with a developmental delay of 4 weeks [7]. Recently, we developed an innovative hip-knee-ankle interlimb-coordinated robot interactive gait training (RIGT) system (P&S Mechanics, Seoul, Korea). The RIGT system was designed to provide accurate proprioceptive, kinematic, and kinetic guidance, and virtual reality (VR) and augmented reality (AR) interactive exercises during gait training [9,10,11]. Compared with current exoskeletal-robot-assisted gait-training systems, the RIGT system can be utilized for sensorimotor feedback (visual and auditory) during locomotor training. Furthermore, separated ankle-knee-hip joint actuators can be provided to ankle dorsiflexion. This prevents foot drop and associated asymmetrical circumduction, thereby enabling safer ambulation and minimizing the risk of falling. Accumulating evidence suggests that VR-AR is effective in increasing the patient’s level of motivation and interest, which can enhance engagement in therapeutic environments [12,13,14]. In the present study, we used VR-AR to facilitate accurate gait training in adolescents with AS, to prevent the risk of falls and muscle imbalance associated with reduced concentration. Therefore, the purpose of this case study was to investigate the effects of RIGT on gross motor function, risk of falls, and kinetic (force) and kinematic (joint angle) locomotor coordination in an adolescent with AS. We hypothesized that RIGT combined with VR-AR intervention would produce better outcomes, in terms of gross motor function, risk of falls, and kinetic and kinematic measures in an adolescent with AS.

## 2. Materials and Methods

### 2.1. Case Description

The participant (Kim) was a 15-year-old male with AS. The study was approved by the Yonsei University Mirae campus Institutional Review Board (1041849-202108-BM-126-02). Initially, the participant was wheelchair-bound and referred to our center to improve ambulation and cognitive and psychological (aggressive and impulsive) behaviors. At the start of the intervention, he mostly used a wheelchair. The participant had been receiving conventional physical therapy and sensory integration therapy prior to participation in the present case study. However, according to the caregiver interview, the patient experienced balance and falling dysfunction. The primary goal at the time of examination was to improve confidence in reducing fall likelihood and improving balance while walking to ease the burden on his family. The participant’s parents provided informed consent prior to beginning the study. Demographic and clinical characteristics are shown in Table 1.

### 2.2. Robotic Interactive Gait-Training System

The RIGT system was designed to facilitate earlier ambulation by allowing the user to adjust body-weight-bearing control and real-time visual biofeedback for torque and stiffness. Kinematics for the hip, knee, and ankle joints provide accurate ankle-knee-hip joint movement and facilitate earlier ambulation by providing adjusted body-weight-bearing control and real-time visual biofeedback for torque and stiffness, and kinematics for the hip, knee, and ankle joints. The device consists of a suspension harness for body-weight support, a motorized treadmill, and a hip-knee-ankle coordinated exoskeleton. The treadmill speed and torque, assistance force, and resistive force can be adjusted to accommodate the patient’s level of locomotor performance as the training progresses [15] (Figure 1).

The RIGT system provides weight-bearing control and real-time audiovisual feedback for torque, active, and resistive force, and kinematics for the hip and knee joints, and provides VR games and AR scenes to maximize the user motivation and concentration [9,10] (Figure 2A,B).

### 2.3. Experimental Task and Procedures

RIGT was performed two times per week for 10 weeks (20 sessions) for 30 min per session on the Walkbot (P&S mechanics, Seoul, Korea) system, excluding the time required for set-up. Time for breaks was provided when requested by the participant; however, the treatment time was maintained for at least 30 min. The participant’s anthropometric data, including age, height, weight, hip-knee length, knee-ankle length, and foot length were initially measured and used for RIGT training. He was equipped with harness and exoskeletal attachments for the thigh, knee, and foot, which were adapted on the basis of anthropometric characteristics. Gait parameter data, including velocity (1.1 km/h), 0% body-weight suspension, and duration and frequency of walking sessions (30 min per day per week) were documented. The participant successfully completed 20 sessions of intervention training [16]. RIGT speed and torque (passive and active) were adjusted to accommodate the participant’s locomotor performance as the training progressed. The session incorporated VR-AR games (e.g., a virtual side-scrolling game—Jordon jumping and taking the coins) and AR scenes (e.g., three-dimensional walking to explore a king’s castle) to maximize patient motivation and concentration [9,17,18,19,20,21], while decreasing anxiety and depression. Clinical outcome and biomechanical measures were obtained.

### 2.4. Clinical Outcome Measurements

#### 2.4.1. Tinetti Performance-Oriented Mobility Assessment

The performance-oriented mobility assessment (POMA) test was used to measure dynamic and static balance and gait [22]. The POMA version used in this clinical evaluation consisted of eight balance items and eight gait items. Balance test items included assessment of sitting balance, rising from a chair and sitting down again, standing balance (eyes open and eyes closed), and turning balance; a maximum score of 12 points can be achieved [23]. The gait test items included assessment of gait initiation, step length, step height, step length symmetry and continuity, path direction, and trunk sway; a maximum score of 16 points can be achieved. The total score ranges from 0 to 28 points [24].

#### 2.4.2. Gross Motor Function Measures

This GMFM test was specifically developed for clinical and research purposes and is used to assess five domains of gross motor function in children with disabilities [25]. The five areas include A (lying and rolling), B (sitting), C (crawling and kneeling), D (standing), and E (walking, running, and jumping). It contains 66 categories. Each item has a score of 3, with 17 items for area A, 20 for B, 14 for C, 13 for D, and 24 for E. The score obtained in each domain divided by the possible score ×100 is the score for each region, and the addition of each domain score divided by 5 becomes the total score [26].

#### 2.4.3. Pediatric Balance Scale

The pediatric balance scale (PBS) is used to assess functional sitting and standing balance ability in children with neuromuscular motor impairments [27]. Fourteen test items are included as follows: moving from a seated to standing position, moving from a standing position to sitting position, transfer, standing without support, sitting without support, standing with eyes closed, standing with feet together, standing with one foot in front of the other, standing on one foot, rotating 360 degrees, turning to look back, picking up an object off the floor, placing alternate foot on step or footrest, and reaching forward with an extended arm. The test score ranges from 0 (“low function”) to 4 (“highest function”), with a total possible score of 56 points [28].

#### 2.4.4. Short Fall Efficacy Scale

The short falls efficacy scale (sFES) questionnaire is used to measure fear of falling in patients with neuromuscular motor impairments during a range of activities of daily living included in the activity domain of the international classification of functioning, disability and health (ICF) model. The sFES comprises seven items. Each activity of daily living is scored on a 4-point scale as follows: 1 (“not at all concerned”), 2 (“somewhat concerned”), 3 (“fairly concerned”), and 4 (“very concerned”). Total possible scores range from 7 (“low concern”) to 28 (“high concern”) [21,29].

#### 2.4.5. Biomechanical Measurement

Biomechanical measurement was used to determine joint movement and force information during treadmill gait before and after Walkbot robotic gait training. The participant underwent biomechanical measurements, including hip and knee joint kinematics (angular displacement) and kinetics (torque, active, and resistive force). The RIGT system is used in conjunction with gait evaluation mechanics software (GEMS), including kinematic and kinetic computing software, which can calculate angular displacement, moment or torque of the hip, knee, and ankle joints in real time. The GEMS was determined using the kinematic and kinetic computing software (P&S Mechanics, Seoul, Korea) of the RIGT system, which calculates joint angular displacement, active and resistive hip, knee, and ankle joint forces, and torque. Kinematic measurements encompassed the joint angle, angular velocity, and acceleration, which were then used to calculate the moment or torque associated with the active and resistive forces of the body segment acting on the ankle, knee, and hip joints of the participant during walking [15]. Kinetic measurements included the active and resistive forces and torques of the body segment acting on the hip joint during RIGT. With the thigh lever arm acting on the RIGT system, the recorded force data can be converted into hip joint torques acting between the RIGT system and the participant’s leg. The hip-knee-ankle joint torque data were collected by the servomotors mounted in the robotic system, in which the corresponding encoders modulate the hip, knee, and ankle joint kinetics. Specifically, active force is defined as a positive directional rotation force that occurs in line with the target movement direction, whereas the resistive force is defined as a negative directional rotation force that acts against the target movement direction [30,31]. The RIGT system is modulated by six servomotors for the bilateral hip, knee, and ankle joints, which enable the safe, coordinated control of locomotor kinematics and kinetics using encoders and a full dynamic model of the exoskeletal system. Each actuated servomotor has a built-in encoder to instantaneously detect the joint angle, angular velocity, and acceleration, and is used for torque computation. Foot load sensors are used to monitor the pressure distribution of each foot when the exoskeleton’s feet are in contact with the treadmill surface, thus providing information on the center of the pressure trajectories during the stance phase. All foot sensors are hardwired to a network circuit of electronic input/output module boards, which are then linked to a central command and operating system. The validity and feasibility of intelligent RIGT systems are well established [30,32] (Table 2).

## 3. Results

### 3.1. Clincial Outcome Measurements

The POMA score increased from 12 (pre-robotic walking) to 15 (post-robotic walking), which was a 25% improvement in gait and balance ability. The GMFM score increased from 43 (pre-robotic training) to 48 (post-robotic training), which was an 11.62% increase in gross motor function. The PBS score increased from 1 (pre-robotic walking) to 32 (post-robotic walking), indicating a 3100% improvement in functional balance ability. The sFES score increased from 16 (pre-robotic walking) to 19 (post-robotic walking), indicating a 18.75% improvement in fear of falling (Table 3).

### 3.2. Biomechanics Measurements

Both knee and hip joint torque values increased approximately 20% as a function of robotic training, supporting clinical improvement in the active muscle strength required for gait function. The resistive hip joint torque decreased; however, the kinematic analysis revealed a significant increase in maximal hip flexion angle after Walkbot training.

## 4. Discussion

The present case study is the first clinical evidence highlighting the positive effects of RIGT combined with VR-AR on gait ability, as determined by POMA score, gross motor function by GMFM score, balance by PBS score, knee and hip joint kinetics, and kinematics during gait in an adolescent patient with AS. As hypothesized, all outcome measures related to balance, gait, and biomechanical characteristics (hip and knee active torque, active and resistive force, and joint angular displacement kinematics) were enhanced after 20 sessions of intensive RIGT combined with VR-AR. To the best of our knowledge, no previously published studies are available for comparison with our RIGT case study.

Clinical balance analysis demonstrated that RIGT effectively increased POMA (25%) and PBS (3100%) in an adolescent patient with AS. This finding was consistent with that of previous gait and balance ability results [33,34,35]. Cesar et al. (2020) reported that eight weeks of gait training with a motor-assisted elliptical device improved PBS (17.0%) and timed up-and-go (16.4%) in 13 children with cerebral palsy and autism [33]. Yazici et al. (2019) found that robotic gait training improved balance (2.5%) more than standard physical therapy after 12 weeks in 24 children with hemiparetic cerebral palsy (CP) [34]. Sukal-Moulton et al. (2014) found an increase in PBS (8.5%) after 6 weeks with the application of a robotic ankle training system in 28 children with CP [35]. A possible explanation for this is that the RIGT provides an ample number of repetitions, with accurate proprioceptive and kinesthetic pressure sense on the hip, knee, and ankle joints, resulting in neuroplasticity and associated functional balance motor recovery [17].

Analysis of gross motor function measures demonstrated that RIGT effectively increased GMFM (11.6%) in an adolescent with AS. The present findings on GMFM were consistent with those of previous robotic-gait-training studies [16,36]. Recently, Jin et al. (2020) showed more enhanced GMFM-88 (34.7%) after 6 weeks of RIGT application in 20 children with CP who were able to ambulate with assistance [16]. Wallard et al. (2018) reported that 20 sessions of robotic-assisted gait training (RAGT) using the hip-knee modulated RAGT (Lokomat pediatric) improved GMFM (13.4%) in 30 children with CP [36]. A possible explanation could be that cortical reorganization enhanced by task-specific training is related to the intensity and frequency of RIGT training [37]. Impaired selective motor control occurs when abnormal flexor or extensor synergies interfere with ankle joint movements, resulting in impaired gait and gross motor movement. It has been suggested that adolescents with CP consistently present with a lack of isolated knee extensor (e.g., quadriceps) control and ankle plantarflexion (e.g., gastrocnemius) control. This finding suggests that RIGT may have facilitated ankle proprioception and kinesthesia, which plays an important role in “reference correction” during locomotor re-training. A possible underlying rationale is that corrective ankle guidance during RIGT-enhanced ankle joint movement sensation and awareness is required for selective, coordinated neuromotor control of active tibialis anterior facilitation and reciprocal inhibition of GCM during gait training. Furthermore, kinetic and kinematic biomechanical analyses demonstrated an improvement of approximately 20–25.2% in the hip and knee joint force and angular movement performance test during gait, as a function of RIGT in the present case. It is possible that the Walkbot RAGT accurately guided the interlimb hip-knee-ankle movement in a coordinated fashion to improve the participant’s gait pattern while improving the motivation and interest with AR and VR games, which corroborates previous results [9,16].

Some research limitations should be considered for future studies. One major limitation in the present case study was the limited sample size. Although we demonstrated positive therapeutic effects of RIGT combined with VR-AR in a participant with AS, caution should be exercised when interpreting our results. Please note that we were unable to implement a clinical study with a large sample size, as AS is an extremely rare genetic condition in children. Another limitation was that our robotic system was a stationary exoskeletal robotic-gait-training device, which allowed safe and repetitive gait training. However, utilizing a wearable robot may better assist community-based ambulation training. Nevertheless, a wearable system is not safe for children with cognitive impairments and may compromise safety in children with AS.

## 5. Conclusions

The present case study demonstrated that RIGT combined with VR and AR can improve gait ability, as demonstrated by POMA score, gross motor function by GMFM score, balance by PBS score, knee and hip joint kinetics, and kinematics during gait in an adolescent patient with AS. Our clinical and biomechanical evidence provide important clinical insights to improve the effectiveness of current neurorehabilitation approaches for treating patients with AS, in balance and locomotor control dysfunction and reduction in the fear of falling.

## Figures and Tables

**Figure 1 children-09-00544-f001:**
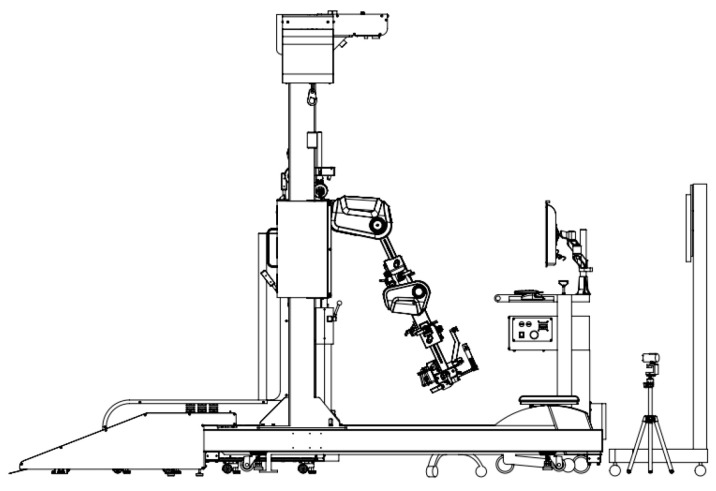
Robot interactive gait training Walkbot.

**Figure 2 children-09-00544-f002:**
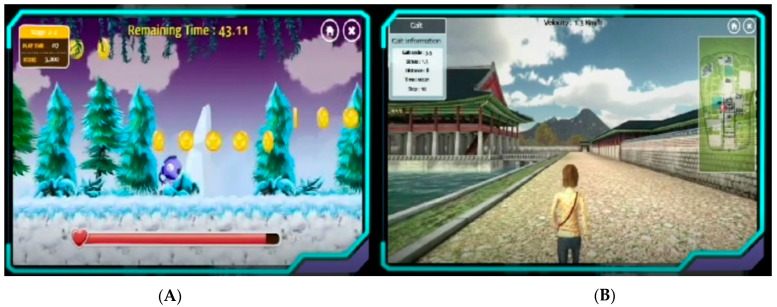
(**A**). Side-scrolling virtual reality game in Walkbot. (**B**). 3D walking-exploring the King’s castle augmented reality in Walkbot.

**Table 1 children-09-00544-t001:** Clinical characteristics of the adolescent.

Sex	Male
Age (years)	15
Height (cm)	148
Weight (kg)	43
GMFCS ^1^ level	II
CARS ^2^	38 (moderate autism)
FIM ^3^ cognitive section	17/35

^1^ GMFCS, Gross Motor Function Classification System; ^2^ CARS, Child Autism Rating Scale; ^3^ FIM, Functional Independence Measure.

**Table 2 children-09-00544-t002:** Clinical outcome measurements.

	Pretest	Post-Test
POMA ^1^	12	15
GMFM ^2^	43	48
PBS ^3^	1	32
sFES ^4^	16	19

^1^ POMA, Performance-Oriented Mobility Assessment; ^2^ GMFCS, Gross Motor Function Classification System; ^3^ PBS, Pediatric Balance Scale; ^4^ sFES, Short Falls Efficacy Scale.

**Table 3 children-09-00544-t003:** Biomechanics measurements.

Hip	Pre-Test	Post-Test
Joint torque (Lt ^1^/Rt. ^2^)	8.80/6.10	10.20/12.30
Active force (Lt./Rt.)	3.23/2.81	5.67/8.30
Resistive force (Lt./Rt.)	25.31/22.60	17.76/16.31
Joint kinematics (Lt./Rt.)	10.69/10.20	14.44/11.76
Knee		
Joint torque (Lt./Rt.)	8.10/9.30	10.40/12.30
Active force (Lt./Rt.)	6.56/7.10	11.42/16.32
Resistive force (Lt./Rt.)	11.70/10.80	6.50/8.50
Joint kinematics (Lt./Rt.)	21.56/29.40	28.33/30.45

^1^ Lt, Left; ^2^ Rt, Right.

## Data Availability

The data presented in this study are available on request from the corresponding author.

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
