# Peer review of "Effects of Robotic Interactive Gait Training Combined with Virtual Reality and Augmented Reality on Balance, Gross Motor Function, Gait Kinetic, and Kinematic Characteristics in Angelman Syndrome: A Case Report"

_children, 2022, doi:10.3390/children9040544_

Round 1
Reviewer 1 Report
This is a well wrigten case report showing the positive results of combining RIGT withVR and AR. I ask them tl provide some specific clarification, as follows. Lines 191-192. The authors should clarify why an increase in PBS from 1 to 32 means a 97% improvement in functional balance abiti. Furthermore, they write in line 211 that PBS increased of 310%. They need clarifyiing such apparent inconsistencies.
Line 227. The acronym RAGT appears here for the first time. The authors should clarify what they mean.
Author Response
This is a well written case report showing the positive results of combining RIGT with VR and AR. I ask them it provide some specific clarification, as follows. Lines 191-192. The authors should clarify why an increase in PBS from 1 to 32 means a 97% improvement in functional balance abilities. Furthermore, they write in line 211 that PBS increase of 310%. They need clarifying such apparent inconsistencies.
Authors response: This was revised (Line 204 and 224).
Line 227. The acronym RAGT appears here for the first time. The authors should clarify what they mean.
Authors response: This was revised (Line 240).

Reviewer 2 Report
Thank you for this excellent and well-written case report, which applies to patient care. The field of robotics and VR is an exciting field. As far as limitations are concerned, if in future there would be a case series, what can we expect in terms of limitations? Kindly consider addressing explicit bias of SMART in this technology.
In Methods/Design can we translate this to a layperson to reach a broader audience of possibly a parent?
Perhaps explaining to the audience the connection between CP and AS.
Author Response
Thank you for this excellent and well-written case report, which applies to patient care. The field of robotics and VR is an exciting field. As far as limitations are concerned, if in future there would be a case series, what can we expect in terms of limitations? Kindly consider addressing explicit bias of SMART in this technology.
Authors response: This was revised (Lines 259-268).
In Methods/Design can we translate this to a layperson to reach a broader audience of possibly a parent?
Perhaps explaining to the audience, the connection between CP and AS.
Authors response: This was revised in methods section.
